# SING: Symbol-to-Instrument Neural Generator

**Alexandre Défossez**
Facebook AI Research
INRIA / ENS
PSL Research University
Paris, France
defossez@fb.com

**Neil Zeghidour**
Facebook AI Research
LSCP / ENS / EHESS / CNRS
INRIA / PSL Research University
Paris, France
neilz@fb.com

**Nicolas Usunier**
Facebook AI Research
Paris, France
usunier@fb.com

**Léon Bottou**
Facebook AI Research
New York, USA
leonb@fb.com

**Francis Bach**
INRIA
École Normale Supérieure
PSL Research University
francis.bach@ens.fr

## Abstract

Recent progress in deep learning for audio synthesis opens the way to models that directly produce the waveform, shifting away from the traditional paradigm of relying on vocoders or MIDI synthesizers for speech or music generation. Despite their successes, current state-of-the-art neural audio synthesizers such as WaveNet and SampleRNN [24, 17] suffer from prohibitive training and inference times because they are based on autoregressive models that generate audio samples one at a time at a rate of 16kHz. In this work, we study the more computationally efficient alternative of generating the waveform frame-by-frame with large strides. We present SING, a lightweight neural audio synthesizer for the original task of generating musical notes given desired instrument, pitch and velocity. Our model is trained end-to-end to generate notes from nearly 1000 instruments with a single decoder, thanks to a new loss function that minimizes the distances between the log spectrograms of the generated and target waveforms. On the generalization task of synthesizing notes for pairs of pitch and instrument not seen during training, SING produces audio with significantly improved perceptual quality compared to a state-of-the-art autoencoder based on WaveNet [4] as measured by a Mean Opinion Score (MOS), and is about 32 times faster for training and $2,500$ times faster for inference.

## 1 Introduction

The recent progress in deep learning for sequence generation has led to the emergence of audio synthesis systems that directly generate the waveform, reaching state-of-the-art perceptual quality in speech synthesis, and promising results for music generation. This represents a shift of paradigm with respect to approaches that generate sequences of parameters to vocoders in text-to-speech systems [21, 23, 19], or MIDI partition in music generation [8, 3, 10]. A commonality between the state-of-the-art neural audio synthesis models is the use of discretized sample values, so that an audio sample is predicted by a categorical distribution trained with a classification loss [24, 17, 18, 14]. Another significant commonality is the use of autoregressive models that generate samples one-by-one, which leads to prohibitive training and inference times [24, 17], or requires specialized implementations and low-level code optimizations to run in real time [14]. An exception is parallel WaveNet [18] which generates a sequence with a fully convolutional network for faster inference. However, the parallel

approach is trained to reproduce the output of a standard WaveNet, which means that faster inference comes at the cost of increased training time.

In this paper, we study an alternative to both the modeling of audio samples as a categorical distribution and the autoregressive approach. We propose to generate the waveform for entire audio frames of 1024 samples at a time with a large stride, and model audio samples as continuous values. We develop and evaluate this method on the challenging task of generating musical notes based on the desired instrument, pitch, and velocity, using the large-scale NSynth dataset [4]. We obtain a lightweight synthesizer of musical notes composed of a 3-layer RNN with LSTM cells [12] that produces embeddings of audio frames given the desired instrument, pitch, velocity[1] and time index. These embeddings are decoded by a single four-layer convolutional network to generate notes from nearly 1000 instruments, 65 pitches per instrument on average and 5 velocities.

The successful end-to-end training of the synthesizer relies on two ingredients:

- A new loss function which we call the *spectral loss*, which computes the 1-norm between the log power spectrograms of the waveform generated by the model and the target waveform, where the power spectrograms are obtained by the short-time Fourier transform (STFT).

  Log power spectrograms are interesting both because they are related to human perception [6], but more importantly because the entire loss is invariant to the original phase of the signal, which can be arbitrary without audible differences.

- Initialization with a pre-trained autoencoder: a purely convolutional autoencoder architecture on raw waveforms is first trained with the spectral loss. The LSTM is then initialized to reproduce the embeddings given by the encoder, using mean squared error. After initialization, the LSTM and the decoder are fine-tuned together, backpropagating through the spectral loss.

We evaluate our synthesizer on a new task of pitch completion: generating notes for pitches not seen during training. We perform perceptual experiments with human evaluators to aggregate a Mean Opinion Score (MOS) that characterizes the naturalness and appealing of the generated sounds. We also perform ABX tests to measure the relative similarity of the synthesizer's ability to effectively produce a new pitch for a given instrument, see Section 5.3.2. We use a state-of-the-art autoencoder of musical notes based on WaveNet [4] as a baseline neural audio synthesis system. Our synthesizer achieves higher perceptual quality than Wavenet-based autoencoder in terms of MOS and similarity to the ground-truth while being about 32 times faster during training and 2, 500 times for generation.

## 2   Related Work

A large body of work in machine learning for audio synthesis focuses on generating parameters for vocoders in speech processing [21, 23, 19] or musical instrument synthesizers in automatic music composition [8, 3, 10]. Our goal is to learn the synthesizers for musical instruments, so we focus here on methods that generate sound without calling such synthesizers.

A first type of approaches model power spectrograms given by the STFT [4, 9, 25], and generate the waveform through a post-processing that is not part of the training using a phase reconstruction algorithm such as the Griffin-Lim algorithm [7]. The advantage is to focus on a distance between high-level representations that is more relevant perceptually than a regression on the waveform. However, using Griffin-Lim means that the training is not end to end. Indeed the predicted spectrograms may not come from a real signal. In that case, Griffin-Lim performs an orthogonal projection onto the set of valid spectrograms that is not accounted for during training. Notice that our approach with the spectral loss is different: our models directly predict waveforms rather than spectrograms and the spectral loss computes log power spectrograms of these predicted waveforms.

The current state-of-the-art in neural audio synthesis is to generate directly the waveform [24, 17, 19]. Individual audio samples are modeled with a categorical distribution trained with a multiclass cross-entropy loss. Quantization of the 16 bit audio is performed (either linear [17] or with a $\mu$-law companding [24]) to map to a few hundred bins to improve scalability. The generation is still extremely costly; distillation [11] to a faster model has been proposed to reduce inference time at the expense of an even larger training time [18]. The recent proposal of [14] partly solves the issue with

a small loss in accuracy, but it requires heavy low-level code optimization. In contrast, our approach trains and generate waveforms comparably fast with a PyTorch[2] implementation. Our approach is different since we model the waveform as a continuous signal and use the spectral loss between generated and target waveforms and model audio frames of 1024 samples, rather than performing classification on individual samples. The spectral loss we introduce is also different from the power loss regularization of [18], even though both are based on the STFT of the generated and target waveforms. In [18], the primary loss is the classification of individual samples, and their power loss is used to equalize the average amplitude of frequencies over time. Thus the power loss cannot be used alone to learn to reconstruct the waveform.

Works on neural audio synthesis conditioned on symbolic inputs were developed mostly for text-to-speech synthesis [24, 17, 25]. Experiments on generation of musical tracks based on desired properties were described in [24], but no systematic evaluation has been published. The model of [4], which we use as baseline in our experiments on perceptual quality, is an autoencoder of musical notes based on WaveNet [24] that compresses the signal to generate high-level representations that transfer to music classification tasks, but contrarily to our synthesizer, it cannot be used to generate waveforms from desired properties of the instrument, pitch and velocity without some input signal.

The minimization by gradient descent of an objective function based on the power spectrogram has already been applied to the transformation of a white noise waveform into a specific sound texture [2]. However, to the best of our knowledge, such objective functions have not been used in the context of neural audio synthesis.

## 3 The spectral loss for waveform synthesis

Previous work in audio synthesis on the waveform focused on classification losses [17, 24, 4]. However, their computational cost needs to be mitigated by quantization, which inherently limits the resolution of the predictions, and ultimately increasing the number of classes is likely necessary to achieve the optimal accuracy. Our approach directly predicts a single continuous value for each audio sample, and computes distances between waveforms in the domain of power spectra to be invariant to the original phase of the signal. As a baseline, we also consider computing distances between waveforms using plain mean square error (MSE).

### 3.1 Mean square regression on the waveform

The simplest way of measuring the distance between a reconstructed signal $\hat{x}$ and the reference $x$ is to compute the MSE on the waveform directly, that is taking the Euclidean norm between $x$ and $\hat{x}$,

$$\mathrm{L_{wav}}\left(x, \hat{x}\right) := \left\|x - \hat{x}\right\|^2. \tag{3.1}$$

The MSE is most likely not suited as a perceptual distance between waveforms because it is extremely sensitive to a small shift in the signal. Yet, we observed that it was sufficient to learn an autoencoder and use it as a baseline.

### 3.2 Spectral loss

As an alternative to the MSE on the waveform, we suggest taking the Short Term Fourier Transform (STFT) of both $x$ and $\hat{x}$ and compare their absolute values in log scale. We first compute the log spectrogram

$$l(x) := \log\left(\epsilon + \left|\mathrm{STFT}\left[x\right]\right|^2\right). \tag{3.2}$$

The STFT decomposes the original signal $x$ in successive frames of 1024 time steps with a stride of 256 so that a frame overlaps at 75% with the next one. The output for a single frame is given by 513 complex numbers, each representing a specific frequency range. Taking the point-wise absolute values of those numbers represents how much energy is present in a specific frequency range. We observed that our models generated higher quality sounds when trained using a log scale of those coefficients. Previous work has come to the same conclusion [4]. We observed that many entries

of the spectrograms are close to zero and that small errors on those parts can add up to form noisy artifacts. In order to favor sparsity in the spectrogram, we use the $\|\cdot\|_1$ norm instead of the MSE,

$$L_{\mathrm{stft},1}(x, \hat{x}) := \|l(x) - l(\hat{x})\|_1. \tag{3.3}$$

The value of $\epsilon$ controls the trade-off between accurately representing low energy and high energy coefficients in the spectrogram. We found that $\epsilon = 1$ gave the best subjective reconstruction quality.

The STFT is a (complex) convolution operator on the waveform and the squared absolute value of the Fourier coefficients makes the power spectrum differentiable with respect to the generated waveform. Since the generated waveform is itself a differentiable function of the parameters (up to the non-differentiability points of activation functions such as ReLU), the spectral loss (3.3) can be minimized by standard backpropagation. Even though we only consider this spectral loss in our experiments, alternatives to the STFT such as the Wavelet transform also define differentiable loss for suitable wavelets.

### 3.2.1 Non unicity of the waveform representation

To illustrate the importance of the spectral loss instead of a waveform loss, let us now consider a problem that arises when generating notes in the test set. Let us assume one of the instrument is a pure sinuoide. For a given pitch at a frequency $f$, the audio signal is $x_i = \sin(2\pi i \frac{f}{16000} + \phi)$. Our perception of the signal is not affected by the choice of $\phi \in [0, 2\pi[$, and the power spectrogram of $x$ is also unaltered. When recording an acoustic instrument, the value of $\phi$ depends on any number of variables characterizing the physical system that generated the sound and there is no guarantee that $\phi$ stays constant when playing the same note again. For a synthetic sound, $\phi$ also depends on implementation details of the software generating the sound.

For a sound that is not in the training set and as far as the model is concerned, $\phi$ is a random variable that can take any value in the range $[0, 2\pi[$. As a result, $x_0$ is unpredictable in the range $[-1, 1]$, and the mean square error between the generated signal and the ground truth is uninformative. Even on the training dataset, the model has to use extra resources to remember the value of $\phi$ for each pitch. We believe that this phenomenon is the reason why training the synthesizer using the MSE on the waveform leads to worse reconstruction performance, even though this loss is sufficient in the context of auto-encoding (see Section 5.2). The spectral loss solves this issue since the model is free to choose a single canonical value for $\phi$.

However, one should note that the spectral loss is permissive, in the sense that it does not penalize phase inconsitencies of the complex phase across the different frames of the STFT, which lead to potential artifacts. In practice, we obtain state of the art results (see Section 5) and we conjecture that thanks to the frame overlap in the STFT, the solution that minimizes the spectral loss will often be phase consistent, which is why Griffin-Lim works resonably well despite sharing the same limitation.

## 4 Model

In this section we introduce the SING architecture. It is composed of two parts: a LSTM based sequence generator whose output is plugged to a decoder that transforms it into a waveform. The model is trained to recover a waveform $x$ sampled at 16,000 Hz from the training set based on the one-hot encoded instrument $I$, pitch $P$ and velocity $V$. The whole architecture is summarized in Figure 1.

### 4.1 LSTM sequence generator

The sequence generator is composed of a 3-layer recurrent neural network with LSTM cells and 1024 hidden units each. Given an example with velocity $V$, instrument $I$ and pitch $P$, we obtain 3 embeddings $(u_V, v_I, w_P) \in \mathbb{R}^2 \times \mathbb{R}^{16} \times \mathbb{R}^8$ from look-up tables that are trained along with the model. Furthermore, the model is provided at each time step with an extra embedding $z_T \in \mathbb{R}^4$ where $T$ is the current time step [22, 5], also obtained from a look-up table that is trained jointly. The input of the LSTM is the concatenation of those four vectors $(u_V, v_I, w_P, z_T)$. Although we first experimented with an autoregressive model where the previous output was concatenated with those embeddings, we achieved much better performance and faster training by feeding the LSTM with only on the 4 vectors $(u_V, v_I, w_P, z_T)$ at each time step. Given those inputs, the recurrent network

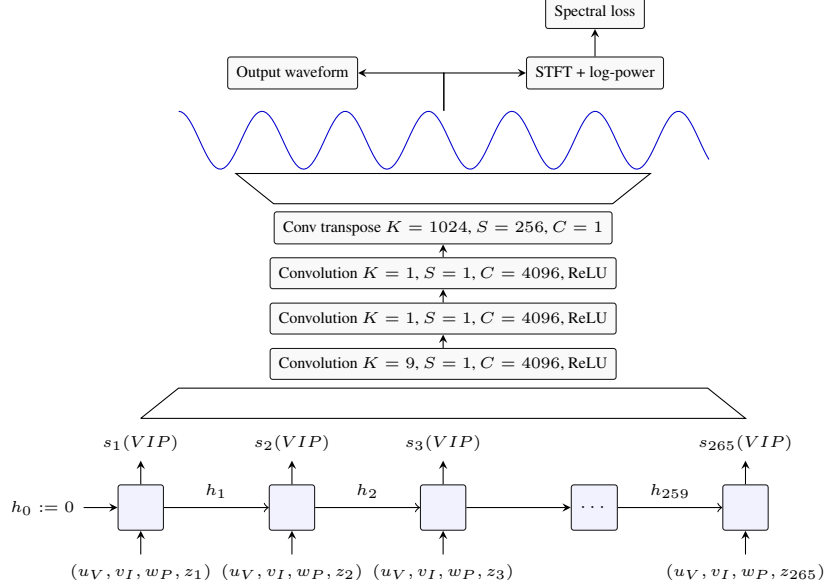

Figure 1: Summary of the entire architecture of SING. $u_V, v_I, w_P, z_*$ represent the look-up tables respectively for the velocity, instrument, pitch and time. $h_*$ represent the hidden state of the LSTM and $s_*$ its output. For convolutional layers, $K$ represents the kernel size, $S$ the stride and $C$ the number of channels.

generates a sequence $\forall 1 \leq T \leq N, s(V, I, P)_T \in \mathbb{R}^D$ with a linear layer on top of the last hidden state. In our experiments, we have $D = 128$ and $N = 265$.

## 4.2 Convolutional decoder

The sequence $s(V, I, P)$ is decoded into a waveform by a convolutional network. The first layer is a convolution with a kernel size of 9 and a stride of 1 over the sequence $s$ with 4096 channels followed by a ReLU. The second and third layers are both convolutions with a kernel size of 1 (a.k.a. 1x1 convolution [4]) also followed by a ReLU. The number of channels is kept at 4096. Finally the last layer is a transposed convolution with a stride of 256 and a kernel size of 1024 that directly outputs the final waveform corresponding to an audio frame of size 1024. In order to reduce artifacts generated by the high stride value, we smooth the deconvolution filters by multiplying them with a squared Hann window. As the stride is one fourth of the kernel size, the squared Hann window has the property that the sum of its values for a given output position is always equal to 1 [7]. Thus the final deconvolution can also be seen as an overlap-add method. We pad the examples so that the final generated audio signal has the right length. Given our parameters, we need $s(V, I, P)$ to be of length $N = 265$ to recover a 4 seconds signal $d(s(V, I, P)) \in \mathbb{R}^{64,000}$.

## 4.3 Training details

All the models are trained on 4 P100 GPUs using Adam [15] with a learning rate of 0.0003 and a batch size of 256.

**Initialization with an autoencoder.** We introduce an encoder turning a waveform $x$ into a sequence $e(x) \in \mathbb{R}^{N \times D}$. This encoder is almost the mirror of the decoder. It starts with a convolution layer with a kernel size of 1024, a stride of 256 and 4096 channels followed by a ReLU. Similarly to the decoder, we smooth its filters using a squared Hann window. Next are two 1x1 convolutions with 4096 channels and ReLU as an activation function. A final 1x1 convolution with no non linearity turns those 4096 channels into the desired sequence with $D$ channels. We first train the encoder and decoder together as an auto-encoder on a reconstruction task. We train the auto-encoder for 50 epochs which takes about 12 hours on 4 GPUs.

**LSTM training.** Once the auto-encoder has converged, we use the encoder to generate a target sequence for the LSTM. We use the MSE between the output $s(V, I, P)$ of the LSTM and the output $e(x)$ of the encoder, only optimizing the LSTM while keeping the encoder constant. The LSTM is trained for 50 epochs using truncated backpropagation through time [26] using a sequence length of 32. This takes about 10 hours on 4 GPUs.

**End-to-end fine tuning.** We then plug the decoder on top of the LSTM and fine tune them together in an end-to-end fashion, directly optimizing for the loss on the waveform, either using the MSE on the waveform or computing the MSE on the log-amplitude spectrograms and back propagating through the STFT. At that point we stop using truncated back propagation through time and directly compute the gradient on the entire sequence. We do so for 20 epochs which takes about 8 hours on 4 GPUs. From start to finish, SING takes about 30 hours on 4 GPUs to train.

Although we could have initialized our LSTM and decoder randomly and trained end-to-end, we did not achieve convergence until we implemented our initialization strategy.

## 5 Experiments

The source code for SING and a pretrained model are available on our github[3]. Audio samples are available on the article webpage[4].

### 5.1 NSynth dataset

The train set from the NSynth dataset [4] is composed of 289,205 audio recordings of instruments, some synthetic and some acoustic. Each recording is 4 second long at 16,000 Hz and is represented by a vector $x_{V,I,P} \in [-1, 1]^{64,000}$ indexed by $V \in \{0, 4\}$ representing the velocity of the note, $I \in \{0, \ldots, 1005\}$ representing the instrument, $P \in \{0, \ldots, 120\}$ representing the pitch. The range of pitches available can vary depending on the instrument but for any combination of $V, I, P$, there is at most a single recording.

We did not make use of the validation or test set from the original NSynth dataset because the instruments had no overlap with the training set. Because we use a look-up table for the instrument embedding, we cannot generate audio for unseen instruments. Instead, we selected for each instrument 10% of the pitches randomly that we moved to a separate test set. Because the pitches are different for each instrument, our model trains on all pitches but not on all combinations of a pitch and an instrument. We can then evaluate the ability of our model to generalize to unseen combinations of instrument and pitch. In the rest of the paper, we refer to this new split of the original train set as the train and test set.

### 5.2 Generalization through pitch completion

We report our results in Table 1. We provided both the performance of the complete model as well as that of the autoencoder used for the initial training of SING. This autoencoder serves as a reference for the maximum quality the model can achieve if the LSTM were to reconstruct perfectly the sequence $e(x)$.

Although using the MSE on the waveform works well as far as the autoencoder is concerned, this loss is hard to optimize for the LSTM. Indeed, the autoencoder has access to the signal it must reconstruct, so that it can easily choose which *representation* of the signal to output as explained in Section 3.2.1. SING must be able to recover that information solely from the embeddings given to it as input. It manages to learn some of it but there is an important drop in quality. Besides, when switching to the test set one can see that the MSE on the waveform increases significantly. As the model has never seen those examples, it has no way of picking the right representation. When using a spectral loss, SING is free to choose a *canonical* representation for the signal it has to reconstruct and it does not have to remember the one that was in the training set. We observe that although we have a drop in

|  |  | Spectral loss | | Wav MSE | |
|---|---|---|---|---|---|
| **Model** | **training loss** | *train* | *test* | *train* | *test* |
| Autoencoder | waveform | 0.026 | 0.028 | 0.0002 | 0.0003 |
| SING | waveform | 0.075 | 0.084 | 0.006 | 0.039 |
| Autoencoder | spectral | 0.028 | 0.032 | N/A | N/A |
| SING | spectral | 0.039 | 0.051 | N/A | N/A |
| SING no time embedding | spectral | 0.050 | 0.063 | N/A | N/A |

Table 1: Results on the train and test set of the pitch completion task for different models. The first column specifies the model, either the autoencoder used for the initial training of the LSTM or the complete SING model with the LSTM and the convolutional decoder. We compare models either trained with a loss on the waveform (see (3.1)) or on the spectrograms (see (3.3)). Finally we also trained a model with no temporal embedding.

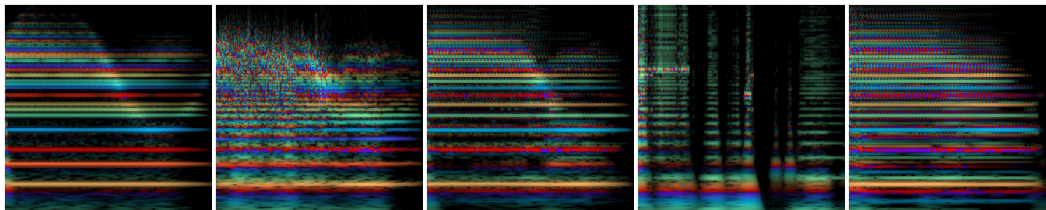

Figure 2: Example of rainbowgrams from the NSynth dataset and the reconstructions by different models. Rainbowgrams are defined in [4] as "a CQT spectrogram with intensity of lines proportional to the log magnitude of the power spectrum and color given by the derivative of the phase". Time is represented on the horizontal axis while frequencies are on the vertical one. From left to right: ground truth, Wavenet-based autoencoder, SING with spectral loss, SING with waveform loss and SING without the time embedding.

quality between the train and test set, our model is still able to generalize to unseen combinations of pitch and instrument.

Finally, we tried training a model without the time embedding $z_T$. Theoretically, the LSTM could do without it by learning to count the number of time steps since the beginning of the sequence. However we do observe a significant drop in performance when removing this embedding, thus motivating our choice.

On Figure 2, we represented the rainbowgrams for a particular example from the test set as well as its reconstruction by the Wavenet-autoencoder, SING trained with the spectral and waveform loss and SING without time embedding. Rainbowgrams are defined in [4] as "a CQT spectrogram with intensity of lines proportional to the log magnitude of the power spectrum and color given by the derivative of the phase". A different derivative of the phase will lead to audible deformations of the target signal. Such modification are not penalized by our spectral loss as explained in Section 3.2.1. Nevertheless, we observe a mostly correct reconstruction of the derivative of the phase using SING. More examples from the test set, including the rainbowgrams and audio files are available on the article webpage[5].

## 5.3 Human evaluations

During training, we use several automatic criteria to evaluate and select our models. These criteria include the MSE on spectrograms, magnitude spectra, or waveform, and other perceptually-motivated metrics such as the Itakura-Saito divergence [13]. However, the correlation of these metrics with

| Model | MOS | Training time (hrs * GPU) | Generation speed | Compression factor | Model size |
|---|---|---|---|---|---|
| Ground Truth | $3.86 \pm 0.24$ | - | - | - | - |
| Wavenet | $2.85 \pm 0.24$ | 3840* | 0.2 sec/sec | 32 | 948 MB |
| SING | $3.55 \pm 0.23$ | 120 | 512 sec/sec | 2133 | 243 MB |

Table 2: Mean Opinion Score (MOS) and computational load of the different models. The training time is expressed in hours * GPU units, the generation time is expressed as the number of seconds of audio that can be generated per second of processing time. The compression factor represents the ratio between the dimensionality of the audio sequences ($64, 000$ values) and either the latent state of Wavenet or the input vectors to SING. We also report the size of the models, in MB.
($*$) Time corrected to account for the difference in FLOPs of the GPUs used.

human perception remains imperfect, this is why we use human judgments as a metric of comparison between SING and the Wavenet baseline from [4].

### 5.3.1  Evaluation of perceptual quality: Mean Opinion Score

The first characteristic that we want to measure from our generated samples is their naturalness: how good they sound to the human ear. To do so, we perform experiments on Amazon Mechanical Turk [1] to get a Mean Opinion Score for the ground truth samples, and for the waveforms generated by SING and the Wavenet baseline. We did not include a Griffin-Lim based baseline as the authors in [4] concluded to the superiority of their Wavenet autoencoder.

We randomly select 100 examples from our test set. For the Wavenet-autoencoder, we pass these 100 examples through the network and retrieve the output. The latter is a pre-trained model provided by the authors of [4][6]. Notice that all of the 100 samples were used for training of the Wavenet-autoencoder, while they were not seen during the training of our models. For SING, we feed it the instrument, pitch and velocity information of each of the 100 samples. Workers are asked to rate the quality of the samples on a scale from 1 ("Very annoying and objectionable distortion. Totally silent audio") to 5 ("Imperceptible distortion"). Each of the 300 samples (100 samples per model) is evaluated by 60 Workers. The quality of the hardware used by Workers being variable, this could impede the interpretability of the results. Thus, we use the crowdMOS toolkit [20] which detects and discards inaccurate scores. This toolkit also allows to only keep the evaluations that are made with headphones (rather than laptop speakers for example), and we choose to do so as good listening conditions are necessary to ensure the validity of our measures. We report the Mean Opinion Score for the ground-truth audio and each of the 2 models in Table 2, along with the $95\%$ confidence interval.

We observe that SING shows a significantly better MOS than the Wavenet-autoencoder baseline despite a compression factor which is 66 times higher. Moreover, to spotlight the benefits of our approach compared to the Wavenet baseline, we also report three metrics to quantify the computational load of the different models. The first metric is the training time, expressed in hours multiplied by the number of GPUs. The authors of [4], mention that their model trains for 10 days on 32 GPUs, which amounts to 7680 hours*GPUs. However, the GPUs used are capable of about half the FLOPs compared to our P100. Therefore, we corrected this value to 3840 hours*GPUs. On the other hand, SING is trained in 30 hours on four P100, which is 32 times faster than Wavenet. A major drawback of autoregressive models such as Wavenet is that the generation process is inherently sequential: generating the sample at time $t + 1$ takes as input the sample at time $t$. We timed the generation using the implementation of the Wavenet-autoencoder provided by the authors, in its *fastgen* version[7] which is significantly faster than the original model. This yields a 22 minutes time to generate a 4-second sample. On a single P100 GPU, Wavenet can generate up to 64 sequences at the same time before reaching the memory limit, which amounts to 0.2 seconds of audio generated per second. On the other hand, SING can generate 512 seconds of audio per second of processing time, and is thus 2500 times faster than Wavenet. Finally, SING is also efficient in memory compared to Wavenet, as the model size in MB is more than 4 times smaller than the baseline.

### 5.3.2 ABX similarity measure

Besides absolute audio quality of the samples, we also want to ensure that when we condition SING on a chosen combination of instrument, pitch and velocity, we generate a relevant audio sample. To do so, we measure how close samples generated by SING are to the ground-truth relatively to the Wavenet baseline. This measure is made by performing ABX [16] experiments: the Worker is given a ground-truth sample as a reference. Then, they are presented with the corresponding samples of SING and Wavenet, in a random order to avoid bias and with the possibility of listening as many times to the samples as necessary. They are asked to pick the sample which is the closest to the reference according to their judgment. We perform this experiment on 100 ABX triplets made from the same data as for the MOS, each triplet being evaluated by 10 Workers. On average over 1000 ABX tests, 69.7% are in favor of SING over Wavenet, which shows a higher similarity between our generated samples and the target musical notes than Wavenet.

## Conclusion

We introduced a simple model architecture, SING, based on LSTM and convolutional layers to generate waveforms. We achieve state-of-the-art results as measured by human evaluation on the NSynth dataset for a fraction of the training and generation cost of existing methods. We introduced a spectral loss on the generated waveform as a way of using time-frequency based metrics without requiring a post-processing step to recover the phase of a power spectrogram. We experimentally validated that SING was able to embed music notes into a small dimension vector space where the pitch, instrument and velocity were disentangled when trained with this spectral loss, as well as synthesizing pairs of instruments and pitches that were not present in the training set. We believe SING opens up new opportunities for lightweight quality audio synthesis with potential applications for speech synthesis and music generation.

### Acknowledgments

Authors thank Adam Polyak for his help on setting up the human evaluations and Axel Roebel for the insightful discussions. We also thank the Magenta team for their inspiring work on NSynth.

## Footnotes

[1]Quoting [4]: "MIDI velocity is similar to volume control and they have a direct relationship. For physical intuition, higher velocity corresponds to pressing a piano key harder."

[2]https://pytorch.org/

[3]`https://github.com/facebookresearch/SING`

[4]`https://research.fb.com/publications/sing-symbol-to-instrument-neural-generator`

[5] https://research.fb.com/publications/sing-symbol-to-instrument-neural-generator

[6] https://github.com/tensorflow/magenta/tree/master/magenta/models/nsynth

[7] https://magenta.tensorflow.org/nsynth-fastgen

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
