[Reviews · NeurIPS 2018]

Reviewer 1



This paper presents an end-to-end audio synthesizer, which generates output at the sample level, while the loss is computed at a frame-level. This alleviates the main problem in neural audio synthesis, which is the exorbitant cost of optimizing sample-based methods. By performing sample generation and model optimization at different rates (16000 vs 100 per second), the proposed approach avoids the main bottleneck, and seems to achieve good results. The paper describes a reasonable set of experiments, although I feel the authors could have provided more analysis of the convergence behavior (is the non-convergence without pre-training a fundamental issue when separating prediction and optimization steps, or is this just a fluke in their particular setup?), and a comparison with Griffin-Lim based methods. The authors write that with Griffin-Lim mapping, artifacts are "likely" to appear, but still these methods work exceedingly well, so clearly there's something there. Still, it is good to have an alternative method like the proposed one. The paper could use some proofreading by a native speaker, although the quality of the writing and presentation is ok in general.

Reviewer 2



Neural audio synthesis is a important and rapidly evolving field of study. Recent breakthroughs rely on expensive audio-sample level autoregressive models. The authors demonstrate a simpler frame-wise regression approach can do better than expected by generating raw audio, but then performing an STFT and regressing to targets in the log-spectral domain. This results in considerable speedups in inference time, as the sound can be generated in parallel. The authors wisely focus on NSynth as a controlled dataset to test conditional generation in a limited domain, and demonstrate generation that generalizes to unseen conditioning, generates audio that is low in distortion, and scores well on MOS tests. Strengths: + STFTs as differentiable operations are a very promising direction. Despite being an obvious choice, sucessful examples of their use are relatively rare, and is the core of the novelty of this paper. + First demonstration of low-distorion audio generated framewise with learned convolutional filters. Framewise filters are usually somewhat noisy when learned to generate raw audio. The combination of the spectral loss and the 'overlap-add' windowing are likely responsible, and demonstrating this is of broad value to those working with machine learning and audio. + Conditional generation of pairs unseen pairs of conditioning demonstrates generalization of the model and the strong use of the conditioning variables. Forcing autoregressive models to utilize conditioning information is sometimes still a challenge as the model can sometimes learn to ignore them and rely on the autoregression for modeling dependencies. Room for improvement: - The paper is a bit confused about its treatement of phase. The model uses a log power spectra as the regression loss function, which is invariant to phase values of local bins. The authors seem to indicate in L45-46 that this is good because audio perception is also invariant to phase, but this is the not the case. Audio perception is very sensitive to phase alignment from frame-to-frame and between harmonics in a single frame. Humans are relatively unaware of *constant phase offsets* applied to the whole spectrogram, which equate to a translation of the audio forward or backwards by a few milliseconds. The log-magnitude loss is also insensitive to constant phase offsets, but gives no information to the model as to how to realistically align the generated phases between frames and frequencies. This is very apparent in the generated audio, as indicated by a percieved lower frequency "warble" or "ringing". Autoregressive models handle the phase problem by working at the lowest possible stride, and it is clear that this model does not comperably handle phase. Section 3.2.1 also misrepresents the situation, by considering an example signal that is a single harmonic and a single frame. Even for a single harmonic, the value of the phase will precess at a constant rate in a stft porportional to the difference of frequency between the fft bin and the harmonic frequency. This is why the derivative of the unwrapped phase for a constant tone is a constant, giving the solid colors in the "Rainbowgrams" of the original NSynth paper. That said, the generated audio sounds much better than magnitudes with random phases. It would improve the paper greatly to more accurately describe the relationships of phase alignment and directly address phase as a limitation of the current technique. For example, presenting rainbowgrams of real, wavenet, and sing generated audio could highlight what aspects sing does well at, despite not receiving any phase error info during training, and what needs improvement. It would be good to include those graphs in the supplemental for all audio in the supplemental. Similarly, a baseline model that just estimates magnitudes for each frame and then uses Griffin-Lim to estimate phase would be very important for arguing that generating directly in the audio domain is an improvement for frame-wise generation. An extension to the current model, using log-magnitude and derivative of unwrapped phase as regression targets would perhaps generate more phase coherent audio. - Reliance on temporal encodings reduces the generality of the proposed architecture. This requires the dataset to be preprocessed and well-aligned, which by design is the case for for the NSynth dataset. - There are a few simple simple metrics that could be added to evaluations to make them more thorough. Since there are labels for the dataset, training a separate classifier could give classification accuracy (pitch, velocity, instrument). The authors could also use the network to compute Frechet Inception Distance (FID), as a proxy for "quality". It would allow for investigation of which pitches/velocities/instruments for which SING does better. Small other comments: * The authors of the NSynth paper actually note that it takes 10 days on 32 K40 GPUs, which are significantly slower than P100 GPUs. (https://github.com/tensorflow/magenta/tree/master/magenta/models/nsynth). While the SING model still is much faster train it's probably more like a factor of 30-40 in reality. Either demphasizing the specific numbers of the cliam (not mentioning in the abstract), or coming up with a more quantitative measure (ex. FLOPS) would be more transparent. * The text of the paragraph on L61 could be clearer. It could just be said that prior work focuses on speech and compositions of symbolic notes, while the current goal is learning timbre of individual notes. * L94, the NSynth model can generate conditional waveforms (pitch, instrument, velocity) since it is an autoencoder (this is how the authors make a playable instrument with it). Also the baseline model generalized to generate pitches that were not seen at training. I think the distinction the authors are trying to make, that should be emphasized is that SING is a generative model, not an autoencoder, so requires no initial audio to condition on. * L137, "spectal loss instead of a regression loss". The spectral loss _is_ a regression loss. I think the authors are trying to distinguish between regression to spectral coefficients and regression to wavefunction values. * L193, "fasten", I think the authors mean to say "speed up" or "hasten" * MOS of random mechanical turkers is perhaps less meaningful than in the case of speech, because while average people are experts at producing and listening to speech, the same cannot be said for music. For example, the phase artifacts present in SING might stand out quite strongly as unrealistic to a music producer or researcher, but less so to a casual listener who might associate "distortion" more with white noise or quantization errors. There's not a lot that the authors can do about that in the context of their experiments, but I just wanted to note, because it was strange to me that the MOS was similar for the SING and wavenet models, while to my ears the wavenet was far preferable most of the time.

Reviewer 3



SUMMARY. The task addressed in this paper is a computationally feasible way to synthesize audio, where the focus is to generalize by interpolating in pitch and velocity space for given instruments. This is a narrower aim than, e.g. the NSynth wavenet-based autoencoder, but it is 85 times faster for training and 2500 times faster for inference. Essential components of the system include a combined convolutional auto-encoder, a 3-layer RNN with LSTM cells, and a loss function that operates in log-spectrogram space. The results are approximately comparable to the wavenet-based autoencoder, although the problem being solved is a different one. QUALITY. Overall, the steps and choices described in the paper all seem quite reasonable, given the problem the authors have set out to address. My disappointment occurred when I listened to the results after having read Section 5.3, which had apparently set my expectations slightly too high. In particular: (1) I found the SING samples to generally be slightly more distorted than those produced by the NSYnth system, and (2) I also found the SING samples to generally be further from the ground truth sample than those produced by the NSynth system. I am well aware that these kinds of samples are very hard to generate, and in particular to do so efficiently, so the fact that they are in the ballpark is itself a success. I am just rather surprised at-- and don't quite understand-- how it is that they were rated more highly--if only slightly--than the NSynth-generated samples. They are not bad at all; they sound to me like they have been passed through a slight ring modulator. It should be further noted that the NSynth samples were ones that the (NSynth) system had seen during training, whereas they had not been seen by the current (SING) system. CLARITY. The paper is written clearly and relatively easy to read. Some typos and a few minor stylistic issues need to be checked. ORIGINALITY. The paper has a number of nice ideas, all revolving around towards making their more efficient synthesizer work, ranging from the basic idea of using an RNN+decoder to generate frames of 1024 at a time, to using a loss function that involves the L1 distance between the STFT of the source and predicted waveform. Furthermore, there are clearly lots of details (described in the paper) required to make this whole system work. SIGNIFICANCE. I hesitate on this. It is quite possible that some of the ideas here will generalize or otherwise be applicable to other contexts, but I am not entirely sure how directly they might translate or extend. If the audio results were excellent, or if the experimental section were impressively thorough and systematic, then this question might not arise for me in the same way, but the slightly distorted audio results, make it, altogether, slightly less than convincing. On the other hand, the speedup is significant, by orders of magnitude! All this leads to some uncertainty for me in my assessment, and thus I lowered my confidence from a 4 to a 3.